# GV-Rep: A Large-Scale Dataset for Genetic Variant Representation Learning

**Zehui Li**[*]
Imperial College London
Vector Institute
zl6222@ic.ac.uk

**Vallijah Subasri**[*]
University Health Network
Hospital for Sick Children
Vector Institute
vallisubasri@gmail.com

**Guy-Bart Stan**
Imperial College London
g.stan@imperial.ac.uk

**Yiren Zhao**
Imperial College London
a.zhao@imperial.ac.uk

**Bo Wang** [†]
University Health Network
University of Toronto
Vector Institute
bowang@vectorinstitute.ai

## Abstract

Genetic variants (GVs) are defined as differences in the DNA sequences among individuals and play a crucial role in diagnosing and treating genetic diseases. The rapid decrease in next generation sequencing cost, analogous to Moore's Law, has led to an exponential increase in the availability of patient-level GV data. This growth poses a challenge for clinicians who must efficiently prioritize patient-specific GVs and integrate them with existing genomic databases to inform patient management. To addressing the interpretation of GVs, genomic foundation models (GFMs) have emerged. However, these models lack standardized performance assessments, leading to considerable variability in model evaluations. This poses the question: *How effectively do deep learning methods classify unknown GVs and align them with clinically-verified GVs?* We argue that representation learning, which transforms raw data into meaningful feature spaces, is an effective approach for addressing both indexing and classification challenges. We introduce a large-scale genetic variant dataset, named GV-Rep, featuring variable-length contexts and detailed annotations, designed for deep learning models to learn GV representations across various traits, diseases, tissue types, and experimental contexts. Our contributions are three-fold: (i) **Construction** of a comprehensive dataset with 7 million records, each labeled with characteristics of the corresponding variants, alongside additional data from 17,548 gene knockout tests across 1,107 cell types, 1,808 variant combinations, and 156 unique clinically-verified GVs from real-world patients. (ii) **Analysis** of the structure and properties of the dataset. (iii) **Experimentation** of the dataset with pre-trained genomic foundation models (GFMs). The results highlight a significant disparity between the current capabilities of GFMs and the accurate representation of GVs. We hope this dataset will advance genomic deep learning to bridge this gap.

38th Conference on Neural Information Processing Systems (NeurIPS 2024) Track on Datasets and Benchmarks.

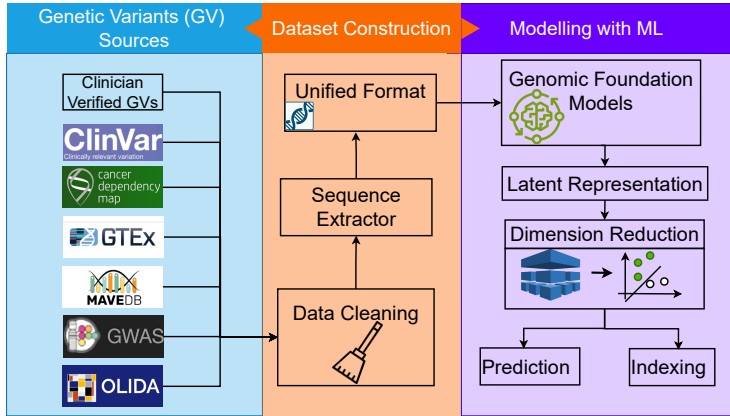

Figure 1: **Overview of the proposed dataset pipeline** The input includes clinician-verified genetic variants from multiple sources like ClinVar and GTEx. These are processed through data cleaning, sequence extraction, and unified formatting. The resulting data is used in genomic foundation models for various tasks such as prediction and indexing.

# 1   Introduction

Genetic variants (GVs) play a pivotal role in disease diagnostics, phenotyping, risk stratification, and as therapeutic targets in drug design and discovery. The advent of next-generation sequencing (NGS) technologies has markedly increased the availability of GV data. This abundance necessitates advanced computational approaches for variant interpretation, which are crucial for advancing personalized medicine and mitigating clinician burnout [32].

Over the past decade, the ACMG-AMP [33] guidelines have become the standard for interpreting and reporting genetic variants (GVs) in the clinical genetic testing of Mendelian disorders, which are characterized by high penetrance and rarity. However, these guidelines do not account for the complex biological processes governing GVs, ranging from alternative splicing [2] and phenotypic variations [37], to changes in gene expression [25] and impacts on cellular fitness [16].

Recent advancements have seen deep learning models applied to GVs with promising results for variant effect prediction (VEP)  [8, 10, 43]. However, evaluation frameworks employed by these models oversimplify the interpretation of GVs, treating them as binary entities: pathogenic variants leading to genetic diseases or benign variants found in healthy populations. This binary classification fails to account for the complexities of genetic expression, disregarding mechanisms like penetrance and expressivity. Penetrance is the proportion of individuals with a specific genotype who exhibit the associated phenotype [9], whereas expressivity refers to the intensity of a given phenotype. GVs can exhibit varying levels of penetrance and expressivity across different biological contexts (e.g. tissue type, cell type, organism) due to epigenetic and epistatic effects [38]. As a result, individuals with the same genetic condition can experience a diverse spectrum of symptoms, underscoring gaps in current VEP datasets and benchmarks. Moreover, in clinical settings, a more nuanced variant classification system is essential for accurate risk assessment and effective genetic counseling [41].

The development of deep learning approaches for modeling these multifactorial effects of GVs is still in its nascent stages, primarily due to the lack of comprehensive datasets that capture the intricate relationships between GVs and their downstream effects on complex traits. While there are existing datasets for modeling GVs, they often suffer from limitations such as insufficient size, lack of diversity, and non-standardized formats that are not conducive to deep learning applications. It's not clear what pre-training datasets and regimes contribute to improved variant effect prediction (VEP). Our work introduces GV-Rep, a large-scale dataset designed to bridge this gap and foster the next generation of genetic variant analysis tools.

Our paper introduces *GV-Rep*, a large-scale dataset of functionally annotated genomic variants (GVs), which could be used for deep learning models to learn meaningful genomic representations. As illustrated in Figure 1, *GV-Rep* aggregates data from seven leading public GV databases and a

---

*Equal contribution.

†Correspondence should be addressed to: `bowang@vectorinstitute.ai`.

Table 1: **Statistics of databases from which GV-Rep is constructed**

| Database | #Variants | Organism Specificity | Cell/Tissue Specificity | Multi-Variants Interactions | Gene Knock-out | Genotype-phenotype association |
|---|---|---|---|---|---|---|
| ClinVar [24] | 1.7M | × | × | × | × | ✓ |
| Cell Passport [40] | 0.7M | × | ✓ | × | × | × |
| Project Score [12] | 17.5K | × | ✓ | × | ✓ | × |
| GTEx eQTLs [6] | 0.6M | × | ✓ | × | × | ✓ |
| GTEx sQTLs [6] | 1.2M | × | ✓ | × | × | ✓ |
| GWAS [36] | 0.3M | × | × | × | × | ✓ |
| MAVEDB [13] | 3.0M | ✓ | × | × | × | × |
| OLIDA [29] | 1.8K | × | × | ✓ | × | ✓ |
| GV-Rep | 7.5 M | ✓ | ✓ | ✓ | ✓ | ✓ |

clinician-validated set compiled by our team. The dataset organizes GV records into a standardized format, consisting of a (reference, alternative, annotation) triplet, and each record is tagged with a label that denotes attributes like pathogenicity, gene expression influence, or cell fitness impact. These annotated records are utilized to fine-tune genomic foundation models (GFMs) [10, 30, 43]. These finetuned GFMs generates meaningful vectorized representations, enabling the training of smaller models for classifying unknown GVs or for search and indexing within a vectorized space.

Our contribution is three-fold:

**Dataset for GV representation learning** We assemble a large-scale GV dataset with more than 7.5 million GV records with diverse labels. This includes 155 well-labeled clinician verified GV records, serving as the anchor GVs for clinical usage (Section 5.3.1).

**Analysis of dataset** We conduct detailed analyses of the *GV-Rep* dataset, examining the distribution and statistics of the variants and labels, highlighting the diversity and unique properties of the dataset.

**Experimentation** We finetune several GFMs with our dataset for classification and indexing. While GFMs achieved more than 65% AUROC in conventional pathogenicity classification, their performance was only marginally better than random chance in more challenging scenarios, such as predicting cell-specific regulation of gene expression with splicing variants. We hope that this work will inspire further research into the representational learning of genetic variations.

The code and dataset are available at `https://github.com/bowang-lab/genomic-FM`.

## 2   Preliminaries and Related Work

**Deep Learning for Genetic Variants** Genetic variants (GVs) are defined as differences observed between an individual's genome and the reference genome. Typically, a genetic variant is represented by a triplet consisting of: chromosome, position, reference nucleotides, and alternative nucleotides. GVs can include single nucleotide variants (SNVs), insertions or deletions (indels), and structural variants, depending on the specific changes in nucleotides [17]. One of the earliest use cases of pathogenicity classification is utilized in a study where CNN models were leveraged to distinguish disease-causing mutations from benign genetic variants [42]. Another study employed a CNN-based architecture to predict the influence of GVs on gene expression [1]. Recently, the focus of deep learning applications to genomics has shifted to large foundation models such as DNABERT2 [43], Nucleotide Transformer [10] and HyenaDNA [30].

**Genomic Datasets for Deep Learning** There is a growing number of datasets emerging across a broad spectrum of genomic tasks from variant effect prediction (VEP) to genomic feature prediction. HyenaDNA leverages the Genomic Benchmarks dataset [30], which focuses on genomic feature prediction of enhancers (in humans and drosophila), non-TATA promoters, regulatory regions, and open chromatin regions [19]. DNABERT2 draws on the Genome Understanding Evaluation (GUE) dataset [43], which encompasses genomic feature prediction tasks including promoter prediction, splice site prediction, COVID-19 variant classification, epigenetic marks prediction, and transcription factor binding sites prediction across human and mouse genomes. Genome Understanding and ANnotation in silico Evaluation (GUANinE) [34] is a benchmark dataset that focuses on prediction of functional elements, conservation, and gene expression. BEND [28] utilizes existing datasets to benchmark both transformer-based and state-space models, demonstrating the versatility and

performance of these architectures for VEP and genomic feature prediction of regulatory regions. Nucleotide Transformer [10] performs evaluation across a broad range of tasks including genomic feature prediction of regulatory elements, splice site, and histone mark, and prioritization of GVs.

**Limitations and Opportunities of Existing GV Datasets** Despite their extensive scope, existing genomic variant (GV) datasets often lack sufficient biological and clinical relevance and complexity, and are constrained by limited dataset sizes and fixed, short context lengths. These benchmarks predominantly focus on tasks such as binary classification of pathogenicity and expression quantitative trait loci (eQTLs) [4, 10, 43]. Moreover, the datasets used are generally derived from major GV databases, with varying criteria for selection across different studies. For example, the BEND benchmark distinguishes between pathogenic and benign variants from ClinVar as classified by Ensembl. GPN-MSA [5] evaluates variant effect by contrasting ClinVar pathogenic variants with those from the gnomAD database [7]. Meanwhile, Nucleotide Transformer assesses ClinVar variants deemed likely pathogenic against variants from the 1000 Genomes project with a minor allele frequency (MAF) greater than 5 percent [10]. In this work, our goal is to develop a GV dataset that surpasses the existing benchmarks in scale, diversity and complexity, minimizes selection bias, and provides a unified format that is optimized for consumption by machine learning algorithms.

**Deep Learning Model for Genomic Assay Prediction** Prior to the emergence of GFMs trained on pure DNA sequences with a reconstruction objective, deep learning had already demonstrated success in predicting genome-scale sequencing assays such as CAGE, DNase-seq, and ChIP-seq. In these tasks, DNA sequences are first mapped to high-dimensional vectors, which are subsequently transformed into real-valued arrays representing the assay results. Over the years, a variety of deep learning models have been developed for these predictions. Early models, such as DeepBind [20], Basenji [22], and Basenji2 [21], utilized CNN-based architectures. More recent approaches have combined CNNs with transformers, as seen in models like Enformer [4] and Borzoi [26]. These models are not only effective in assay prediction but can also be applied to Genetic Variant Prediction. By using the predicted values or intermediate vector representations to form a latent space, scores for genetic variants can be directly extracted through dimensionality reduction techniques or a learned linear head. One challenge of training these models are that annotated dna sequence data with track information are needed for training, this potentially pose challenges to scale up the training.

# 3   Dataset

## 3.1   Dataset Overview

As shown in Table 4, we collected the GV-Rep dataset from seven databases of genetic variant (GV) effects studies. These studies cover a wide range of existing GVs and extend beyond conventional binary classification settings that primarily sub-sample from ClinVar [24]. Specifically, we included GV studies from Cell Passport [40] and Score Projects [12], which provide GV effects with cell- and tissue-specific contexts. Additionally, we incorporated data from OLIDA [29], which presents the effects of multiple GVs on diseases. The resulting dataset, GV-Rep, thereby offers a comprehensive and context-rich resource for the analysis of GV effects.

## 3.2   Dataset Construction

**Construction** Figure 2 illustrates the dataset construction process. Firstly, Genetic Variant (GV) records, their annotations, and associated labels are extracted from databases. To enable downstream machine learning models to process these records, a **sequence extractor** is used to convert GV records by locating the position of the GV in the human reference genome. During this process, GVs located on rare contigs—specifically, contigs not included in the GRCh38/hg38 reference genome—are filtered out.

**Usage** The extracted sequence, which has the GV centered in the middle, is of a context length defined by the user, resulting in a (reference sequence, alternate sequences) pair. These paired sequences, along with the annotations, serve as inputs for fine-tuning or inference with GFMs based on user requirements. Additionally, once the GFMs are finetuned, they can be used to vectorize unknown GVs. This allows the use of vector database tools, such as FAISS [11], to search and match unknown GVs with labeled GVs in the database.

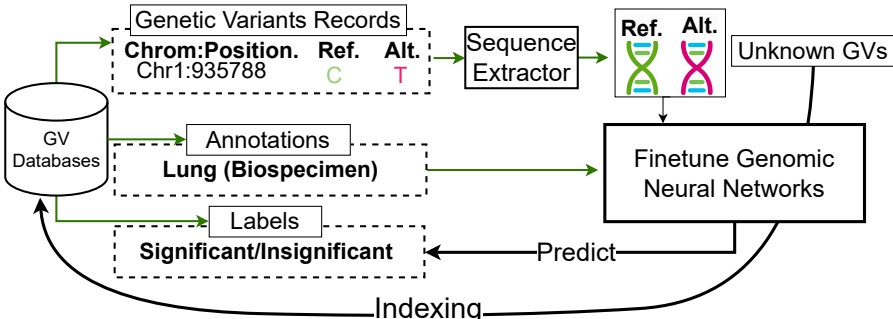

Figure 2: **Dataset Construction and Usage** This diagram give an example on the construction workflow of GV-Rep dataset from a source database. Genetic variant records, containing chromosome position and reference/alternate alleles, along with biospecimen-specific annotations and a binary label indicating the significance of the GV, are extracted from source GV database. The sequence extractor processes these GV records, which can then be used by GFMs for predicting the significance of unknown genetic variants. The finetuned GFMs could encode and index unknown GVs by matching them with GVs in the databases.

**GV Record**: Following this construction process, the minimum unit of GV-Rep dataset is a record, which is an $(x, y)$ pair. Here, $x = (\text{ref}, \text{alt}, \text{annotation})$, and $y$ is the corresponding label indicating the class of GV or a real value quantifying the effects of the GV.

### 3.2.1 Dataset Description

**ClinVar** [24] ClinVar hosts genetic variants (GVs) supported by evidence and classified into four pathogenicity categories across 13,209 disease types: likely benign ($n = 714,866$), benign ($n = 195,030$), pathogenic ($n = 143,348$), and likely pathogenic ($n = 100,859$). The genetic variants in ClinVar can be further classified by variant type as a missense variant, intronic variant, splice donor variant, or synonymous variant.

**Cell Passport** [40] The Cell Model Passports dataset includes curated data on patient samples, model relationships, and over 1,200 established cancer cell lines and organoid models. It provides comprehensive model characteristics, genetic feature summaries, and the capacity to integrate multiple genomic datasets.

**Project Score** [12] This dataset features genome-scale CRISPR-Cas9 drop-out screens across 1,107 cell lines, including extensively annotated cancer models, to identify genes critical for cellular fitness in specific molecular contexts.

**GTEx QTLs** [6] The dataset comprises 1,207,976 expression quantitative trait loci (eQTLs) and 618,932 splicing quantitative trait loci (sQTLs) across 14 tissue types.

**GWAS Catalog** [36] Maintained by NHGRI-EBI, this catalog includes data from over 45,000 genome-wide association studies (GWAS), covering more than 5,000 human traits and hosting over 40,000 datasets with full p-value summary statistics. It features 306,890 SNPs associated with 53,933 traits/diseases that were mapped to their Experimental Factor Ontology (EFO) term [27].

**MAVEDB** [13] This database includes multiplex assays of variant effect (MAVEs), such as deep mutational scans and massively parallel reporter assays. Each experiment tests thousands of variants, providing functional effect scores relative to a reference for genetic elements (e.g. coding sequences, promoters, enhancers). MAVEDB was further curated to include only experiments with complete DNA sequence information, totaling 3,166,541 variants across 1,304 studies.

**OLIDA** [29] The OLIDA database has been curated to include 1,808 high-quality bilocus variant combinations linked to 219 oligogenic diseases as a positive set [29], and 150,500 bilocus combinations from healthy individuals of The Thousand Genomes Project (1KGP) as a negative set [3]. Future models can leverage OLIDA to predict interactions between multiple variants and their combined effects on phenotypes.

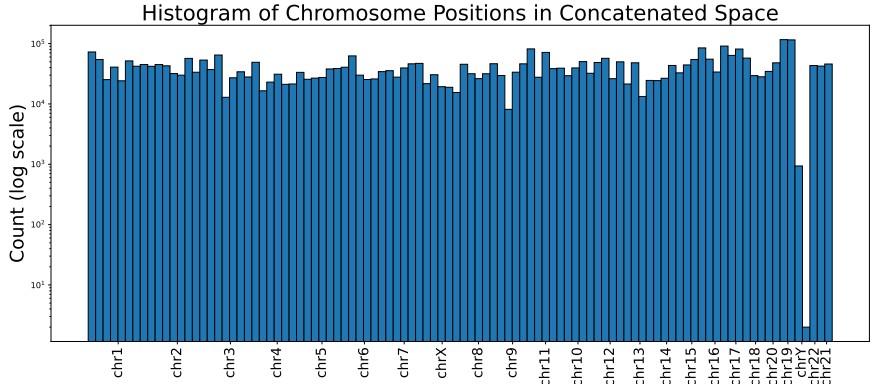

Figure 3: **Distributions of Genetic Variants by Chromosome**. The distribution of GVs are relatively uniform across various chromosomes.

# 4 Dataset Statistics and Analysis

**Basic Statistics** Overall, we have GV-Rep $\mathbb{D} = (\mathcal{X}, \mathcal{Y}) = \{(x^{(n)}, y^{(n)})\}_{n=1}^{N}$, with $N = 28,363,315$ (See Table 4 in Appendix A for breakdown statistics of each type of records). The number of Cell Type Specific Gene-KO records is significantly larger than the other types because for each type of cells, we will have numerous gene-ko experiment, and overall, there are 17,548 gene-ko $\times$ 1107 cell lines records.

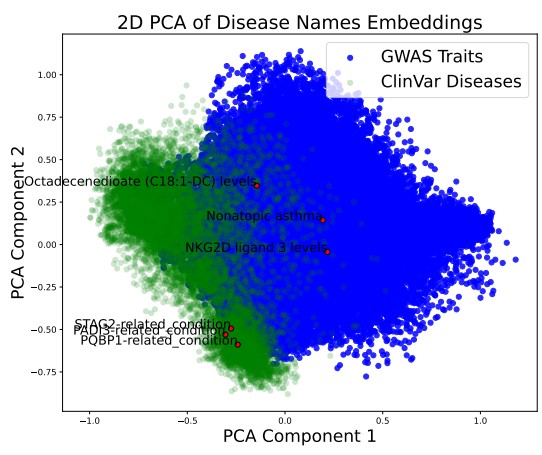

(a) **Distributions of Diseases and Trait Labels**. The disease and trait names are encoded with T5 and projects to the PCA dimensions.

(b) Fitness Score Distributions with Gene-KO. A skewness in the distribution towards the negative side indicates the effectiveness of the gene-KO.

Figure 4: (a) Distributions of Diseases and Trait Labels (b) Gene- KO Fitness Score Distributions

**Variants Distribution and Label Diversity** Figure 3 shows the *distribution of the GV across all the chromosomes*, we can see that GV-Rep covers most of the positions of human chromosomes uniformly. The exception is ChrY, which has much fewer GVs compared to the other chromosomes.

In addition, the labels associated with each GV exhibit significant diversity. **(1) Diseases Coverage**: Our dataset includes 65,153 diseases and disease-related traits sourced from ClinVar and GWAS. Figure 4a illustrates the text embeddings of these diseases and traits, generated using the T5 text encoder [31]. Notably, GWAS traits [36] extend beyond simple disease names, including additional terms such as the expression levels of proteins, ligands, and hormones. In contrast, the disease names

from ClinVar [24] are primarily symptom-focused. **(2) Gene-KO Fitness Score Distribution**: The fitness score describes the influence of knockout of a gene on the host cell. A negative score indicates a statistically significant effect on cell fitness. Figure 4 shows the distribution of fitness scores across 1,107 cell lines and 17,548 genes. The overall distribution skews towards negative values, indicating that most gene knockouts influence the biological activity of genes. **(3) Multiplex Assays of Variant Effect (MAVE) Distribution**: The MAVE score is a normalized quantitative measure that indicates how a specific genetic variant affects a biological trait or function. A negative value indicates the pathogenicity of the variant [14]. Figure 7 shows the distribution of MAVE scores. Overall, the scores are symmetrically distributed around zero.

**Statistics of Clinically Verified GVs**    This dataset contains 155 unique variants from 84 anonymized patients with hereditary cancer predisposition that have been interpreted and classified by board-certified clinical molecular geneticists, in accordance with the ACMG-AMP guidelines [39]. The variants have been prioritized from highest known cancer predisposition potential to lowest using the Cancer Variant Classification Schema by leveraging tiered cancer gene lists, pedigree-based analyses and expert curation. The Cancer Variant Classification Schema consists of five classes that account for (i) the relationship between a given gene and the cancer developed, (ii) the functional consequence of a variant in a particular gene and iii) knowledge of co-segregation and familial inheritance patterns, whereby:

*Class 1:* P/LP variant in a known autosomal dominant cancer predisposition gene (CPG).

*Class 2:* P/LP variant in a known autosomal recessive CPG.

*Class 3:* P/LP variant in a known cancer gene frequently mutated in the somatic context.

*Class 4:* P/LP variant in a novel, candidate cancer gene supported by sufficient evidence.

*Class 5:* Cancer-segregating variants of uncertain significance (VUS) in a known cancer gene.

## 5 Experiment with Genomic Foundation Models

### 5.1 Experiment Setup

To demonstrate the use case of our dataset, we run several experiments with four state-of-the-art pre-trained Genomic Foundation Models (GFMs): HyenaDNA(Hyena) [30], DNABERT2 [43], Nucleotide Transformer (NT), and Nucleotide Transformer v2 (NTv2) [10][3]. During the model finetuning process, we attach a three-layer-CNN header to the frozen pretrained GFMs to aggregate information. PyTorch Lightning is used to implement the finetuning and evaluation process. NVIDIA V100 32GB are used for inference and finetuning. A total of 400 GPU hours are approximately used in total. We use Adam Optimizer [23] and set the learning rate to $1e^{-3}$.

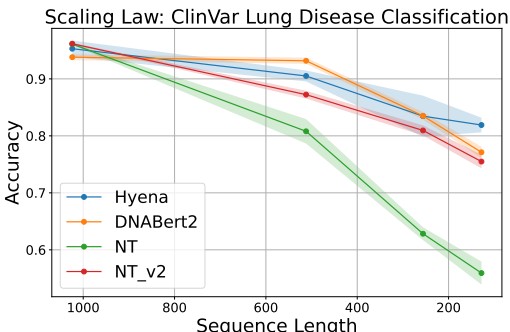

Figure 5: **Scaling Law of Genomic Foundation Models in ClinVar Lung Disease Classification.** The plot shows the accuracy of various models (HyenaDNA, DNABERT2, NT, and NT_v2) vs. sequence length. The context length extends on both sides of the mutated nucleotides of genetic variants.

### 5.2 Variant Property Prediction

**Scaling Law in GV Prediction**    We investigated the influence of context lengths on classification accuracy using the ClinVar disease classification task. Intuitively, the effect of a GV should be context-dependent. Longer contexts should facilitate better prediction of GV effects by modelling long-range interactions. For example, if a GV occurs in a gene-encoding region, it is more likely to be a pathogenic mutation based on it's context. To test this hypothesis, we constructed a simple task to predict whether a given GV would lead to lung-related diseases[4]. Figure 5 shows the AUC-ROC

---

[3]For the details about the model version please refer to appendix C

[4]For the list of lung-related diseases, see Appendix B

Table 2: AUROC ↑ and mean square error (MSE ↓) values for various Genomic Foundation Models running on GV-Rep dataset. The tasks are ClinVar Pathogenicity (ClinVar), sQTL Significance Classification (sQTL), and Gene Knock-Out Fitness Score Prediction (Gene-KO).

| Model | ClinVar (AUROC% ↑) | sQTL (AUROC% ↑) | Gene-KO (MSE ↓) |
|---|---|---|---|
| Random | $50.59 \pm 0.10$ | $50.50 \pm 1.07$ | $1.11 \pm 0.09$ |
| Hyena | $65.05 \pm 0.27$ | $52.20 \pm 2.62$ | $1.06 \pm 0.11$ |
| DNABERT2 | $\textbf{73.87} \pm \textbf{0.21}$ | $52.62 \pm 2.98$ | $1.06 \pm 0.11$ |
| NT-Human | $65.75 \pm 0.25$ | $51.33 \pm 5.87$ | $1.06 \pm 0.11$ |
| NT-V2 | $68.73 \pm 0.27$ | $\textbf{54.10} \pm \textbf{1.13}$ | $1.06 \pm 0.11$ |

of fine-tuned GFMs versus sequence length. It is clear that as we decrease the context length, the prediction accuracy drops correspondingly. Moreover, there is a performance difference between the four models: while Nucleotide Transformer version 2 achieves the best performance with a context length of 1024, it is very sensitive to context length, and its accuracy drops steeply with reduced context length. In contrast, DNABERT2 and Hyena tend to be more robust to changes in context length.

**Fine-Grained and Coarse-Grained Tasks**  While existing tasks mainly focus on pathogenicity prediction, our dataset includes GVs from multiple perspectives, enabling the construction of more fine-grained tasks to predict how a GV influences splicing (sQTL) and how a GV is related to a gene fitness score in a given cell type (Gene-KO). Here, we present the evaluation results of four GFMs on three tasks: pathogenicity classification on ClinVar (four-class classification), splicing effect (sQTLs) (two-class classification), and a regression task, gene knock-out fitness score prediction (Gene-KO). The context length for all tasks is set to 1024 base pairs. Note that, compared to prior work, the ClinVar pathogenicity classification task we have here contains pathogenicity classes, longer context and 1.3 million GVs.

As shown in Table 2, the results are mixed. Overall, existing GFMs struggle with cell- and tissue-level tasks that are heavily influenced by complex regulatory mechanisms: Gene-KO and sQTLs. In sQTLs, most models perform only slightly better than random guessing. In the Gene-KO task, three models converged to the same solution, suggesting that the GFMs are unlikely to provide meaningful representations, and instead, the added header learns to encode the gene-KO values. Additionally, we found that multi-species models (DNABERT2 and NT-V2) tend to perform better on the proposed tasks than models (Hyena and HT-Human) trained with only the human reference genome.

## 5.3 Genetic Variants Indexing

Genomic Foundation Models (GFMs) map GVs into a vector space, enabling the use of vector database tools such as faiss [11] to quickly index GVs - match unknown GVs with annotated GVs and quantify the distances between GVs. The **clinician verified GVs (CVGV)** in our dataset could serve as a testbench to show the effectiveness of GFMs when being applied for GV encoding and indexing.

**Approaches** Given a set of unknown GVs $\mathcal{X} = \{x^{(n)}\}_{n=1}^{N}$, and the well annotated GVs set $\mathbb{A} = (\mathcal{X}_A, \mathcal{Y}_A) = \{(x_A^{(n)}, y_A^{(n)})\}_{n=1}^{N_A}$. we could use GFMs as an encoder $\mathbb{E}_\theta$ and a distance function $d$ to form a Query $Q$. The query can be formed in two ways 1) An unknown variant $x_i$ is used as the keyword and search against in the annotated GVs set $\mathbb{A}$. Then the query will be formed as $Q(\mathbb{E}_\theta(x_i), \mathbb{E}_\theta(\mathbb{A}), d)$, returning a list of known GVs which are similar to $x_i$. Such a clinical use case would be to better understand an unknown genetic variant that has been consistently identified in patient samples. In this scenario, clinicians may want to search to see if there are existing GVs that are similar to the unknown GV, such that it can be better categorized. 2) Set the keyword to be an annotated $x_A^i$, and search against an unknown set of GVs $\mathcal{X}$. This can then be used by genetic counsellors to prioritize unknown GVs from a patient. Here the query will be $Q(\mathbb{E}_\theta(x_A^i), \mathbb{E}_\theta(\mathcal{X}), d)$.

### 5.3.1 Comparison of Indexing Accuracy between Original and Finetuned GFMs

**Task** In CVGV, each variant is annotated with a unique label, indicating to what extent a GV is associated with cancer predisposition. Here we compare how the original pretrained GFMs and finetuned GFMs with 1.3 million ClinVar GVs perform on GV indexing.

**Metric** We first check when querying with a GV with class label k, the optimal indexing algorithm should return GVs which are similar to the query and hereby most of the labels would be k. Here, we sample query vectors and search with the CVGV set, and for each query, we will take the top 10 results and count the number of returned GVs with the same label as the query. The percentage of the GVs with the same label as the query GV is used as the metric, indicating the *Variants Indexing Accuracy*.

Table 3: **Variants Indexing Accuracy of GFMs** on clinician verified GVs with and without (W/O) Finetuning

| Model | *W/O* Finetuning | Finetuned |
|---|---|---|
| Random | $0.334 \pm 0.04$ | - |
| Hyena | $0.428 \pm 0.09$ | $\mathbf{0.662 \pm 0.16}$ |
| DNABERT2 | $0.446 \pm 0.06$ | $0.616 \pm 0.06$ |
| NT | $\mathbf{0.558 \pm 0.10}$ | $0.620 \pm 0.10$ |
| NTV2 | $0.478 \pm 0.06$ | $0.542 \pm 0.05$ |

**Result** As shown in Table 3, the fine-tuned GFMs consistently perform better than their pretrained counterparts across four GFMs. In terms of the difference in model performance, NT achieved the highest accuracy among models without finetuning, while Hyena achieves the best performance after finetuning with 1.3 million pathogenicity records in ClinVar.

### 5.3.2 Querying Time

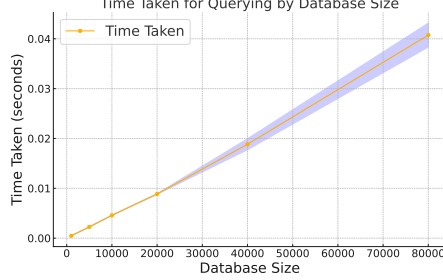
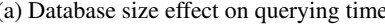
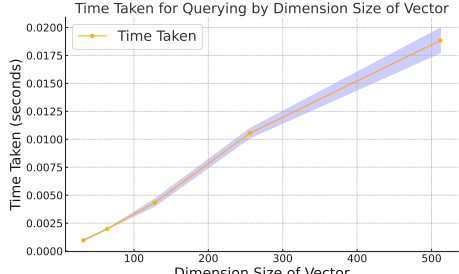

(a) Database size effect on querying time                (b) Vector dimension effect on querying time

Figure 6: (a) Querying time as a function of database size, with vector dimension fixed at 512. (b) Querying time as a function of vector dimension, with database size fixed at 40,000.

In many resource-limited scenarios such as clinical settings, the speed of search operations is critical. Consider a query $Q(\mathbb{E}_\theta(x_i), \mathbb{E}_\theta(\mathcal{X}), d)$, where $\mathbb{E}_\theta$ encodes a genetic variant (GV) into an $m$-dimensional vector, and $\mathcal{X}$ contains $n$ data points. The dimensions of the embedding, $m$, and the number of data points, $n$, significantly influence the speed of the query. This theoretical understanding aligns with the empirical data presented in the subsequent figures.

Figure 6a shows that the querying time increases linearly with the database size, demonstrating that larger databases require more time for data retrieval. Similarly, Figure 6b reveals a linear increase in querying time as the vector dimension grows, suggesting that more complex data representations extend the retrieval process. These observations emphasize the need to carefully consider database size and vector dimensionality to maintain efficient querying performance in practical implementations.

## 6 Limitations and Future Work

While GV-Rep is unlikely to have a negative societal impact, caution must be exercised whenever this dataset is used to inform clinical decision-making. Applications developed using this dataset should be implemented with care to uphold the principle of human-in-the-loop for real-world applications. It is crucial that all genetic variants predicted by machine learning models are thoroughly reviewed and validated by geneticists and clinicians to ensure accurate and responsible use in informing patient care in clinical settings. The use of genetic data also presents risks to the privacy of an individual,

which can be used for genetic discrimination. GV-Rep will be monitored and validated as more data is integrated, ensuring that the dataset continues to reflect the diversity of populations it aims to serve and minimizing the risk of bias over time. This vigilance will help ensure that GV-Rep remains a valuable tool for advancing personalized medicine, while safeguarding against unintended consequences.

Despite the advances offered by the GV-Rep dataset in GV benchmarking, there are several areas ripe for enhancement. Future considerations include leveraging fine-mapping tools that integrate GWAS/QTL association signal strength and linkage disequilibrium (LD) information to prioritize potential causal variants. To improve the fairness and applicability of these models, the dataset's scope should be broadened to include sensitive attributes such as ethnicity and sex. This would facilitate an evaluation of model fairness across demographics, especially for groups typically underrepresented in existing genomic databases. Integrating epigenetic data, like DNA methylation patterns, could deepen our understanding of how these factors influence gene expression and the resulting phenotypic manifestations of genetic variants. Expanding to include longer genetic contexts and cross-species pre-training could provide valuable evolutionary insights and accelerate translational research from model organisms to patient care.

Conducting a thorough assessment of tokenization, pre-training data, and training regimes across a wide range of tasks will inform more effective model training strategies. Additionally, detailed evaluations of variant indexing and scaling laws across various diseases contexts will inform model generalization. Future research should explore diverse fine-tuning approaches to enhance model accuracy and adaptability. Further development of polygenic risk scores and control vectors could lead to more personalized therapeutic strategies, dramatically increasing the dataset's utility and impact on genomics and personalized medicine. Addressing these limitations and focusing on these future directions will be crucial for enhancing the dataset's utility and impact on deep learning applications for GVs.

## Acknowledgements

The authors gratefully acknowledge the funding from research grants provided through the federal Pan-Canadian Artificial Intelligence Strategy at Vector Institute. Zehui Li is grateful to his family—Jinghai Li, Huiqing Wang, and Yiqiu Sun for their support. Valli is grateful for the constant support from all her family and friends, especially RJ.

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

## A  Additional Statistics

## B  List of Diseases

The following list constitutes the superset of lung-related diseases used in the Lung Disease classification task in Section 5.2:

- Lung cancer
- EGFR-related lung cancer
- Lung carcinoma
- Autoimmune interstitial lung disease-arthritis syndrome
- Global developmental delay - lung cysts - overgrowth - Wilms tumor syndrome

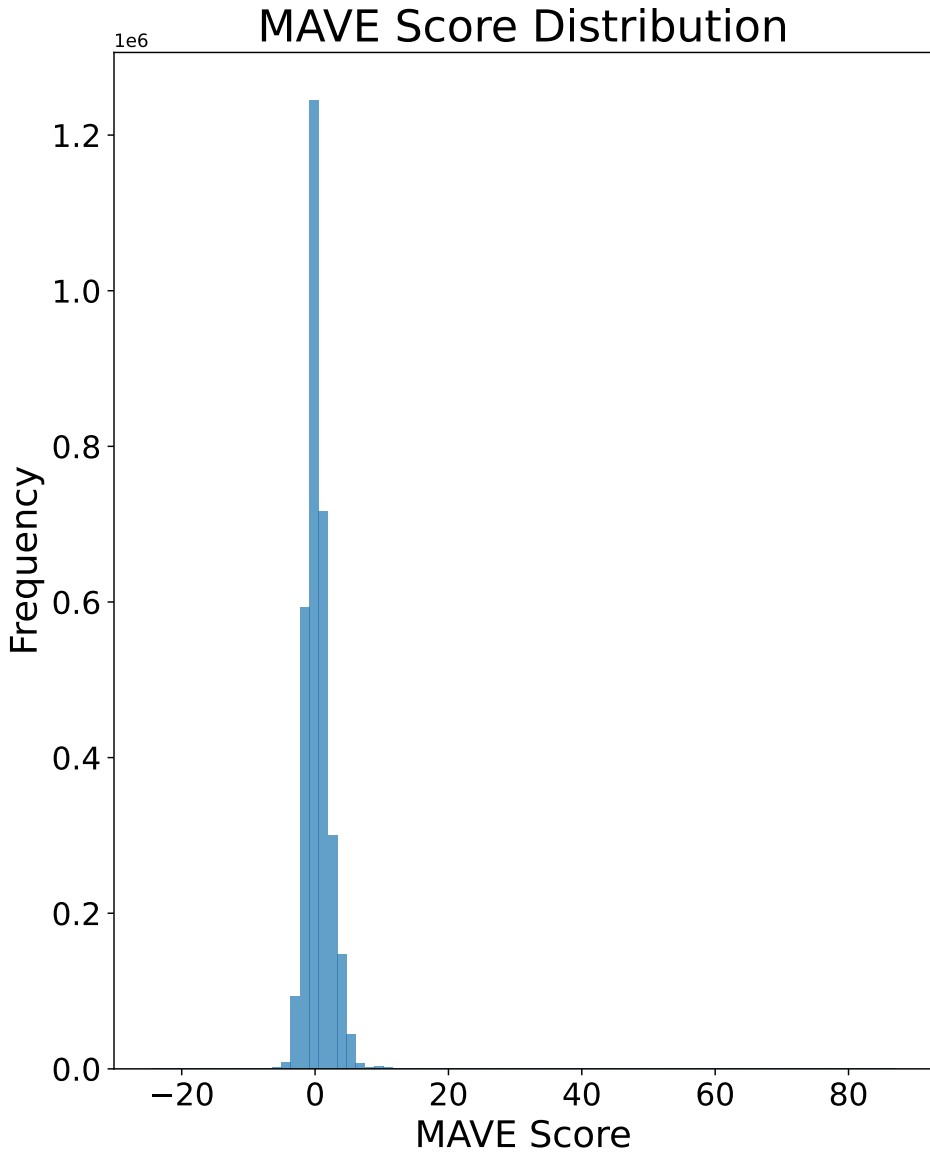

Figure 7: **Distribution of MAVE Scores for Genetic Variants**. Negative values indicate the pathogenicity of the genetic variants (GVs). Overall, the MAVE scores are symmetrically distributed around zero.

Table 4: **Basic Statistics of GV-Rep** We have GV records with both discrete labels (classification) and continuous labels(Predictions).

| Record Types | Train | Valid | Test |
|---|---|---|---|
| # Pathogenicity Classification | 923,282 | 115,410 | 115,411 |
| # Disease Type Classification | 1,391,752 | 173,969 | 173,970 |
| # Trait Type Classification | 245,512 | 30,689 | 30,689 |
| # Organism Specific eQTL Classification | 966,380 | 120,797 | 120,799 |
| # Organism Specific sQTL Classification | 495,145 | 61,893 | 61,893 |
| # Cell Type Specific Cancer Driver Gene Classification | 593,390 | 74,173 | 74,175 |
| # Cell Type Specific Gene Knock-out Effect Prediction | 15,540,508 | 1942563 | 1942565 |
| # Multiplex Assays of Variant Effect Prediction | 2,533,232 | 316,654 | 316,655 |
| # GV-Interaction Effects Classification | 1446 | 180 | 182 |

- Small cell lung carcinoma
- Chronic lung disease
- Lung adenocarcinoma
- Lung disease
- Non-small cell lung carcinoma
- Squamous cell lung carcinoma

## C   Genomic Foundation Models used in Experiments

The following four models are used in Section 5:

- **HyenaDNA**[30]: We use the pretrained hyenadna-tiny-1k model, which can be downloaded from the HuggingFace repository[5].
- **DNABERT2**[43]: We use the available checkpoint from this model.
- **Nucleotide-Transformer-v2** & **Nucleotide-Transformer**: This model, trained on both human genome and multiple species datasets, is available in two versions at HuggingFace: nucleotide-transformer-v2-500m-multi-species[6] and nucleotide-transformer-500m-human-ref[7].

**Implementation Details for Other Models**   We integrate the **BEND** framework [28] (available at `https://github.com/frederikkemarin/BEND/tree/main`), which facilitates the inclusion of other models such as **GENA**[35] and **Grover**[15]. This modular approach allows for easy experimentation and extension with various genomic models.

## D   Ethics Statement

The majority of GV-Rep was generated from public, open-source datasets, whereby the original raw data downloaded from the data sources does not contain any personally identifiable information or sensitive content. For the clinician-verified genetic variant set, patients were consented in accordance with the ethical principles of the Declaration of Helsinki and approved by an Institutional Review Board. All patients were approved for molecular profiling by the SickKids Research Ethics Board (ID: 1000051699). Therefore, we are not aware of any social or ethical concern of GV-Rep. Since GV-Rep

---

[5] `https://huggingface.co/LongSafari`
[6] `https://huggingface.co/InstaDeepAI/nucleotide-transformer-v2-500m-multi-species`
[7] `https://huggingface.co/InstaDeepAI/nucleotide-transformer-500m-human-ref`

is a general dataset for representation learning of GVs, we also do not forsee any direct application of GVs for malicious purposes. However, the users of GVs should be aware of any potential negative social and ethical impacts that may arise from their chosen downstream datasets or tasks outside of GVs, if they intend to use the GVs datasets as pre-training datasets to perform transfer learning.

## E  Licence

GV-Rep is distributed under the CC BY-NC-SA (Attribution-NonCommercial-ShareAlike) license. For the sub-datasets constituting GV-Rep, users are required to follow the guidance and usage policies of the original licenses as specified below. While most sub-datasets are under CC or CC0 licenses, the data from the Cancer Dependency Map should be used according to its original data usage policy for educational purposes only[8].

The Clinician verified GV set is under the CC BY-SA 4.0 license, and the code is under the MIT license[9].

- ClinVar: Creative Commons Public Domain Dedication (CC0 1.0) license
- GTEx: Open Access Data from GTEx is under Creative Commons licenses
- MAVEDB: CC BY-NC-SA 4.0 (Attribution-NonCommercial-ShareAlike)
- GWAS: Creative Commons Public Domain Dedication (CC0 1.0) license
- OLIDA: CC BY-NC-SA 4.0 (Attribution-NonCommercial-ShareAlike)

## F  Code and Data Availability

The code and data are available at `https://github.com/bowang-lab/genomic-FM`.

## G  Dataset Documentation

We following the datasheet from [18] to document our dataset.

### G.1  Motivation

**For what purpose was the dataset created?**

**Who created the dataset (for example, which team, research group) and on behalf of which entity (for example, company, institution, organization)?**  The dataset was created by the author of the paper. ZL, a student at Imperial College London and a visiting scholar at Vector Institute, and VS, a researcher at the Hospital for Sick Children and University Health Network, affiliated with the Vector Institute, were involved in its creation. The project was supervised by BW, who is affiliated with the University Health Network, the University of Toronto, and the Vector Institute.

**Who funded the creation of the dataset?**  Funding for research grants is provided through the federal Pan-Canadian Artificial Intelligence Strategy at Vector Institute.

### G.2  Composition

**What do the instances that comprise the dataset represent (for example, documents, photos, peo- ple, countries)?**  Genetic Variants (GVs) Records and labels describing the effect of GVs.

**How many instances are there in total (of each type, if appropriate)?**  See Table 4 in Appendix A

**Does the dataset contain all possible instances or is it a sample (not necessarily random) of instances from a larger set?**  It is not sampled from a large set, rather it filter out GVs which does not appear on the hg38 chromosomes.

---

[8]https://depmap.sanger.ac.uk/documentation/data-usage-policy/
[9]https://opensource.org/license/mit/

**What data does each instance consist of?** each instance is a $(x, y)$ pair. Here, $x =$ (ref, alt, annotation), and $y$ is the corresponding label indicating the class of GV or a real value quantifying the effects of the GV.

**Is there a label or target associated with each instance?** Yes

**Is any information missing from individual instances?** N/A

**Are relationships between individual instances made explicit (for example, users' movie ratings, social network links)?** N/A

**Are there recommended data splits (for example, training, development/validation, testing)?** We use 0.8:0.1:0.1 by default, but the user can use other split with our code.

**Are there any errors, sources of noise, or redundancies in the dataset?** The GVs could duplicate across different tasks, however, they will be labeled with distinct labels, so the record containing these GVs are not redundant.

**Is the dataset self-contained, or does it link to or otherwise rely on external resources (for example, websites, tweets, other datasets)?** self-contained.

**Does the dataset contain data that might be considered confidential (for example, data that is protected by legal privilege or by doctor–patient confidentiality, data that includes the content of individuals' non-public communications)?** The original clinician-verified data is confidential but the anonymized version of this data is not confidential and can be distributed under the C BY-NC-SA 4.0 licence.

**Does the dataset contain data that, if viewed directly, might be offensive, insulting, threatening, or might otherwise cause anxiety?** No

**Does the dataset identify any subpopulations (for example, by age, gender)?** No

**Is it possible to identify individuals (that is, one or more natural persons), either directly or indirectly (that is, in combination with other data) from the dataset?** No

**Does the dataset contain data that might be considered sensitive in any way (for example, data that reveals race or ethnic origins, sexual orientations, religious beliefs, political opinions or union memberships, or locations; financial or health data; biometric or genetic data; forms of government identification, such as social security numbers; criminal history)?** No. All the GVs are anonymized.

### G.3 Collection process

**What mechanisms or procedures were used to collect the data (for example, hardware apparatuses or sensors, manual human curation, software programs, software APIs)?** Software mainly. See Section 3.2.

**If the dataset is a sample from a larger set, what was the sampling strategy (for example, deterministic, probabilistic with specific sampling probabilities)?** NO

**Who was involved in the data collection process (for example, students, crowdworkers, contractors) and how were they compensated (for example, how much were crowdworkers paid)?** N/A

**Over what timeframe was the data collected?** The data was collected between Feb. 2024 to May 2024. For individual GVs, it has associated timestamp.

**Were any ethical review processes conducted (for example, by an institutional review board)?**
N/A

**Did you collect the data from the individuals in question directly, or obtain it via third parties or other sources (for example, websites)?**   Mainly from third party public websites and a subset of the dataset is collected from the clinicians.

**Were the individuals in question notified about the data collection?**   N/A

**Did the individuals in question consent to the collection and use of their data?**   N/A

**Has an analysis of the potential impact of the dataset and its use on data subjects (for example, a data protection impact analysis) been conducted?**   N/A

### G.4   Preprocessing/cleaning/labeling

**Was any preprocessing/cleaning/labeling of the data done (for example, discretization or bucketing, tokenization, part-of-speech tagging, SIFT feature extraction, removal of instances, processing of missing values)?**   Yes. See Section 3.2

**Was the "raw" data saved in addition to the preprocessed/cleaned/ labeled data (for example, to support unanticipated future uses)?**   Yes. By default it will be saved to `/root/data`

**Is the software that was used to preprocess/clean/label the data available?**   Yes.  All the preprocessing code is available.

### G.5   Uses

**Has the dataset been used for any tasks already?**   No

**Is there a repository that links to any or all papers or systems that use the dataset?**   Yes. See Appendix F

**What (other) tasks could the dataset be used for?**   Any tasks which involving the representation learning of genetic variant will be able to take advantage of this dataset.

**Is there anything about the composition of the dataset or the way it was collected and preprocessed/ cleaned/labeled that might impact future uses?**   N/A

**Are there tasks for which the dataset should not be used?**   See Appendix D

### G.6   Distribution

**Will the dataset be distributed to third parties outside of the entity (for example, company, institution, organization) on behalf of which the dataset was created?**   Yes.

**How will the dataset be distributed (for example, tarball on website, API, GitHub)?**   Github

**When will the dataset be distributed?**   The dataset is now public accessable

**Will the dataset be distributed under a copyright or other intellectual property (IP) license, and/or under applicable terms of use (ToU)?**   See Appendix E

**Have any third parties imposed IP-based or other restrictions on the data associated with the instances?**   See Appendix E

**Do any export controls or other regulatory restrictions apply to the dataset or to individual instances?**   No

## G.7 Maintenance

**Who will be supporting/hosting/maintaining the dataset?**  The authors

**How can the owner/curator/ manager of the dataset be contacted (for example, email address)?** Emails

**Is there an erratum?**  N/A

**Will the dataset be updated (for example, to correct labeling errors, add new instances, delete instances)?**  Yes, the dataset will be updated with new support for other genomic foundation models.

**If the dataset relates to people, are there applicable limits on the retention of the data associated with the instances (for example, were the individuals in question told that their data would be retained for a fixed period of time and then deleted)?**  N/A

**Will older versions of the dataset continue to be supported/hosted/ maintained?**  Yes, the follow-up update will mostly be the addition of other data sources.

**If others want to extend/augment/build on/contribute to the dataset, is there a mechanism for them to do so?**  Yes, we recommend issuing the pull request through Github.

