# OpenReview forum: "GV-Rep: A Large-Scale Dataset for Genetic Variant Representation Learning"
_NeurIPS.cc/2024/Datasets_and_Benchmarks_Track — NeurIPS 2024 Track Datasets and Benchmarks Poster_

### Official Review · Reviewer_3LTZ · 2024-06-17
**A large-scale data resource for variant effect prediction**

**Rating:** 5
**Confidence:** 4
**Correctness:** Claims look adequate and dataset cons…
**Clarity:** The paper is clear.

**Review:**

GV-Rep provides an original large-scale resource of annotated genomic variants. The paper is mostly very clear, and the underlying data is of an adequate quality.

Pro
+ large scale
+ high diversity of variant-based tasks
Con
- limited distinction between coding/noncoding variants

**Strengths:**

GFMs are an increasingly active field, and having access to more diverse variant-based tasks will be useful to many studies going forward.

**Additional Feedback:**

GV-Rep seems relevant to VEP beyond the (narrow) scope of probing DNA LMs/ genome FMs. As so far GFMs have not really been shown to be a winning recipe for VEP, I wonder whether it could not be presented as a general benchmark data resource for VEP ML (Personally I would e.g. not see Enformer fall under the GFM term).

**Documentation:**

Dataset documentation is provided in the supplement. It would be good to provide explicit statistics on the organism distribution for the relevant record types, and discriminate between coding and non-coding variants whereever feasible.

**Ethics:**

No concerns

**Limitations:**

Limitations are addressed.

**Opportunities For Improvement:**

1) Penetrance: Does it make sense to assume that such a population-wide statistic should be predictable using sequence-based models, as suggested by the introduction?
2) Related work: Shouldn't Enformer/Borzoi - style approaches be mentioned explicitly here? They don't explicitly learn from genetic variants, but are used for their interpretation.
3) Dataset Description: it would be good to have more details on the usage of sources like Cell Passport and Score Project, that are not explicitly focused on documenting variants directly. I don't directly understand in what form Score Project provides SNVs.
4) MAVEDB: I understand that the selection criterion includes studies on protein coding variants, rather than experiments focused on the actual DNA. What is the reason for that?
5) Along similar lines, it looks like also for ClinVar no distinction is being made between coding and noncoding variants. Why is a more fine-grained classification, especially fo the human genome, not considered?
6) VEP has often been explored as a zero-shot task. Here a supervised training approach is adopted. Why is zero-shot prediction not considered?
7) Line 228: Why are Gene-KO and sQTLs referred to as epigenetic tasks? I would typically expect things like chromatin or methylation under this term.
8) FAISS-based GV matching. This is very interesting, but as it is a novel idea, baselines would help. Could this also be done using sequence alignment-based search?
9) HyenaDNA: why was tiny1k, the smallest available model, chosen?
10) The OLIDA dataset is introduced but not used any further. Any pointers for how such an interaction dataset should be leveraged for benchmarking using GFMs?

**Relation To Prior Work:**

Related benchmarking work for DNA representations is discussed adequately.

**Summary And Contributions:**

The paper introduces GV-Rep, a benchmark set for probing understanding of genetic variants. A dataset with 7 million variants is curated with the aim of going beyond binary classification of variants.

---

> ### Author Rebuttal · Authors · 2024-08-15
>
> We appreciated your feedback on this paper. Following your suggestion, we have provided the **extended related worked** which will be included in the final draft and also implemented a **filter function** for the user to get protein-encoding vs non-encoding region attribute for any genetic variants. Below, we provide detailed responses to each of your questions. Due to the length limit of the rebuttal, the response to Q2 (extended related work), and Q8 (Sequence alignment-based search as the baseline), Q9 (why was hyenatiny1k is chosen), and Q10 (OLIDA dataset details) are provided as the official comments.
>
> ### Q1. Penetrance
>
> Predicting population-wide penetrance statistics using sequence-based models alone is inherently challenging, as penetrance is shaped by a complex interplay of factors beyond genetic sequence, including environmental influences, epigenetic modifications, and gene interactions. We discuss this in our paper on lines 43-45: “GVs can exhibit varying levels of penetrance and expressivity across different biological contexts (e.g. tissue type, cell type, organism) due to epigenetic and epistatic effects.” While these models can offer valuable insights into the likelihood of specific genetic variants manifesting in phenotypic traits, we are not suggesting that nucleotide sequence data alone can fully account for penetrance, as this would oversimplify the intricate nature of genetic expression. Nevertheless, genetic variants play a crucial role in penetrance, and it is essential to thoroughly assess the performance of sequence-based models in isolation before integrating additional modalities.
>
>
>
> ### Q3. Data format of Cell Passport and Score Project dataset
>
> As detailed in the line 131 and 135 of the original paper, the cell line specific VCF files are available from Cell Passport and Score Projects. Cell Passport provides SNVs (single nucleotide variants) from whole genome sequencing  across a diverse range of cell lines, enabling us to examine the impact of genetic variation within specific cellular contexts. Specificially, VCF data of Cell Passport Project were downloaded from https://cog.sanger.ac.uk/cmp/download/mutations_wgs_vcf_20221123.zip.  The gene-KO raw data are obtained from https://cog.sanger.ac.uk/cmp/download/Project_Score2_fitness_scores_Sanger_v2_Broad_21Q2_20240111.zip, which contains integrated cancer dependency datasets processed as described in Pacini et al., Cancer Cell 2024. These datasets are then processed by the dataset preprocessing functions in the src/datasets folder of the repository.
>
>
>
> ### Q4. Clarification on MAVEDB
>
> Multiplexed assays of variant effects (MAVEs) enable the simultaneous assessment of the functional impacts of thousands of genetic variants. These assays are utilized to derive fitness scores for both non-coding and protein-coding regions within the genome. Each MAVE, depending on its sequence type and study completeness, may include a nucleotide and/or amino acid sequence. In the GV-Rep project, we focused on utilizing nucleotide sequences along with their associated fitness scores, excluding entries that contained only protein data. For more details, you can view a related discussion here: https://youtu.be/BXGQ2IuDnGE?t=1324.
>
>
> ### Q5. Coding and non-coding variants in ClinVar
> Following your suggestions, we have added a filter function to differentiate between coding and non-coding variants to the sequence extractor, this change has been refected in our [code repository](https://github.com/bowang-lab/genomic-FM/pull/52). Now the sequence extractor will optionally provide whether a given genetic variants is within protein-encoding region by refering back to the human genome reference GTF file. This classes can be applied to all the subdatasets, including ClinVar.
>
> Furthermore, when it comes to ClinVar, our dataset loader have more fine grained metadata available for the user to use. The load_clinvar function in the code base will return all the metadata associated with each GVs, this includes the classes of the variants such as **missense_variant**, **intron_variant, splice_donor_variant**, **synonymous_variant**. The user will be available to perform filter to obtain a subset of the dataset they are interested.
>
> ### Q6. Why is zero-shot prediction not considered?
> While the binary classification task can be obtained by the setting a predefined threshold, for the multi-class classification task, the linear probing is needed. In section 5, the ClinVar classification task is a four classes classfication to differentiate between ikely benign, benign, pathogenic, and likely pathogenic. Furthermore, we think adding a probing head to the model can help to generate more expressive vector representation of the DNA sequences. As shown in the Table 3 of Section 3.5.1, we have included the zero-shot performance of four models in the indexing task, where the finetuned model has much better performance.
>
> ### Q7. Why are Gene-KO and sQTLs referred to as epigenetic tasks?
>
> Gene knockouts (Gene-KOs) directly alter DNA by removing or deactivating genes to explore their functions, classifying them as genetic modifications. Similarly, sQTLs (splicing quantitative trait loci), while influencing gene expression, do so by affecting transcriptional mechanisms or mRNA processing, which are fundamentally genetic processes, not epigenetic ones like DNA methylation or histone modifications.
>
> The term "epigenetic-level" on line 228 was intended to convey that these tasks, despite being genetic in nature, can have downstream effects on epigenetic regulation and exhibit significant tissue and cell-type specificity. However, we acknowledge that this terminology could be misleading and appreciate your feedback. In the final draft of the paper, we will replace "epigenetic-level" with the more accurate term "fine-grained" to better reflect the nuanced nature of these tasks.

---

> > ### Author Response · Authors · 2024-08-15
> > **Extended Response to Q2, Q8, Q9, and Q10**
> >
> > ### Q2. Extended related work including Enformer/Borzoi
> > We are aware of Enformer/Borzoi and other deep neural netowrks trained with transcriptomic data. We will add the following paragraph to the extended related work section of Appendix A.
> >
> > > **Deep Learning Model for Genomic Assay Prediction** Prior to the emergence of GFMs trained on pure DNA sequences with a reconstruction objective, deep learning had already demonstrated success in predicting genome-scale sequencing assays such as CAGE, DNase-seq, and ChIP-seq. In these tasks, DNA sequences are first mapped to high-dimensional vectors, which are subsequently transformed into real-valued arrays representing the assay results. Over the years, a variety of deep learning models have been developed for these predictions.
> > Early models, such as DeepBind[1], Basenji[2], and Basenji2[3], utilized CNN-based architectures. More recent approaches have combined CNNs with transformers, as seen in models like Enformer[4] and Borzoi[5]. These models are not only effective in assay prediction but can also be applied to Genetic Variant Prediction. By using the predicted values or intermediate vector representations to form a latent space, scores for genetic variants can be directly extracted through dimensionality reduction techniques or a learned linear head. One challenge of training these models are that annotated dna sequence data with track information are needed for training, this potentially pose challenges to scale up the training.
> >
> >
> > ### Q8. Sequence alignment-based search
> >
> > We initially explored sequence-based alignment with BLASTN as a baseline. However, its direct application to GV indexing proved challenging. While BLASTN is adept at finding local sequence similarities, our task requires a more nuanced understanding of the relationship between the reference and alternate alleles within a GV pair.
> >
> > Simply searching the ALT sequence with BLASTN may miss crucial context, especially for short ALTs or those with low similarity to the reference. Additionally, BLASTN's low-complexity filtering, designed to reduce noise in general sequence searches, could inadvertently obscure true variant signals in our specific context.
> >
> > In contrast, our embedding-based method generates a unified vector representation for the entire (Ref, ALT) pair, encapsulating the complex interplay between the two alleles. This enables efficient indexing and retrieval of GVs based on their inherent semantic similarity, even when the ALT sequence alone exhibits limited homology.
> >
> >
> > ### Q9. HyenaDNA: why was tiny1k, the smallest available model, chosen?
> > As previously stated, we are not aiming to cover all different versions of the model as the focus of this paper is on the dataset side, we will anticipate that larger models will help with the performance. To verify this, we include additional experiments. The below table shows the performance of the Hyena-Middle and Hyena-Large in the lung-related-disease classification task (section 5.1). The context length of three models are set to 1024. It shows that larger models generally have better performance.
> >
> > | Model         | Performance (avg ± std)       |
> > |---------------|------------------------------|
> > | Hyena-Tiny    | 0.9529 ± 0.020      |
> > | Hyena-Middle   | 0.9830 ± 0.001       |
> > | Hyena-Large   | 0.9831 ± 0.000       |
> >
> >
> > ### Q10. OLIDA dataset details
> >
> > The OLIDA dataset, which focuses on variant interactions in oligogenic diseases, was introduced as part of GV-Rep and is accessible from our code repo and data repo.  Future work could leverage OLIDA by incorporating it into oligogenic analysis tasks, where GFMs could be benchmarked on their ability to predict interactions between multiple variants and their combined effects on phenotypes. GV-Rep currently focuses on digenic variant combinations and outputs sequences (Ref, ALT) for both GV pairs. However, this capability opens avenues for adapting GFM architectures to handle the complexity of oligogenic diseases. For instance, multi-input layers could be incorporated into GFMs, enabling them to concurrently process information about multiple alleles and their interdependencies.
> >
> > [1]	Alipanahi, B., et al., 2015. Predicting the sequence specificities of DNA-and RNA-binding proteins by deep learning. Nature Biotechnology, 33(8), pp.831-838.
> >
> > [2]	Kelley, D.R., et al., 2018. Sequential regulatory activity prediction across chromosomes with convolutional neural networks. Genome Research, 28(5), pp.739-750.
> >
> > [3]	Kelley, D.R., 2020. Cross-species regulatory sequence activity prediction. PLoS Computational Biology, 16(7), p.e1008050.
> >
> > [4]	Linder, J., et al., 2023. Predicting RNA-seq coverage from DNA sequence as a unifying model of gene regulation. Biorxiv, pp.2023-08.
> >
> > [5]	Avsec, Ž., et al., 2021. Effective gene expression prediction from sequence by integrating long-range interactions. Nature Methods, 18(10), pp.1196-1203.

---

> > > ### Comment · Reviewer_3LTZ · 2024-08-20
> > >
> > > Thank you for considering my suggestions!
> > >
> > > **One last comment to MaveDB, as I might have been misunderstood:**
> > >
> > > My point was that what was evaluated in many of the assays is the fitness of a protein, encoded by DNA, rather than the fitness of a functional DNA element itself. For benchmarking, I think it makes sense to differentiate between protein-coding DNA and regulatory/noncoding - as they probe very different domains of a model's understanding. As far as I know MaveDB provides this distinction.
> > >
> > > Our final aim should be that a benchmark like GV-Rep helps us understand a model -  I think that e.g. "This model does better on regulatory DNA than on protein-coding DNA" would be a more helpful conclusion to draw than "This model does well on MaveDB". Does the codebase currently support easily subsetting the data to enable such insights?

---

> > > > ### Author Response · Authors · 2024-08-21
> > > > **Clarification and Demo of subsetting capability of GV-Rep**
> > > >
> > > > Thank you for responding to our rebuttal and providing further clarification. Following your suggestion, we added the [filter function](https://github.com/bowang-lab/genomic-FM/pull/52) to our codebase during the rebuttal, to support subsetting genetic variants across all databases. The implementation relies solely on the position of the genetic variant (GV) to differentiate between coding and non-coding regions. Therefore, this subsetting capability can be applied to genetic variants from most databases and is also applicable to external databases which user may bring in.
> > > >
> > > > ### Filter Function Usage
> > > > The filter function has been integrated into the **GenomeSequenceExtractor** class.
> > > >
> > > > ```python
> > > > from src.sequence_extractor import GenomeSequenceExtractor
> > > > ####################################
> > > > ## The record is returned by dataset loaders such as ClinVar, GWAS, QTLs, etc.
> > > > ####################################
> > > >
> > > > # For example, a missense variant from ClinVar
> > > > gv_record = {'Chromosome': '1',
> > > >              'Position': 69134, 'Reference Base': 'A',
> > > >              'Alternate Base': ['G']}
> > > >  # The context length of the retrieved genetic variant
> > > > SEQUENCE_LENGTH = 20
> > > > genome_extractor = GenomeSequenceExtractor(encoding_region_filter=True)
> > > >
> > > > # Extract sequences
> > > > ref, alt, is_protein_encoding = genome_extractor.extract_sequence_from_record(gv_record, SEQUENCE_LENGTH)
> > > >
> > > > ####################################
> > > > ### Output of GenomeSequenceExtractor
> > > > ####################################
> > > >
> > > > >Reference sequence: 'GATTCTCAGGAACTCCAGAC'
> > > > >Alternate sequence: 'GATTCTCAGGGACTCCAGAC'
> > > > >is_protein_encoding: True
> > > > ```
> > > > The filter function works by comparing the position of a given GV to the [GTF genome-wide annotation files](https://ftp.ebi.ac.uk/pub/databases/gencode/Gencode_human/release_46/) to determine whether the GV is protein-coding or not. One limitation of this filter function is when position of GV is missing, but we think this could be a rare case.

---

> > > > > ### Comment · Reviewer_3LTZ · 2024-08-21
> > > > >
> > > > > I'm actually a bit confused by this answer - I would assume most variants that are in MaveDB to have been measured in some expression system on synthetic DNA constructs, not in an actual genomic locus that would be part of a GTF file?
> > > > >
> > > > > Was a mapping done somehow? I just checked again but couldn't see such a thing in the paper. If data was mapped, are there any implications of doing so? I guess an observation might be put in a flanking sequence context that actually was not there when the measurement was obtained.

---

> > > > > > ### Author Response · Authors · 2024-08-21
> > > > > > **Mapping MaveDB  to Reference Genome**
> > > > > >
> > > > > > I initially thought you were broadly referring to the coding vs. non-coding functionality of the codebase. You are correct that the original MaveDB scoreset data does not have mappings to the reference genome and, as a result, lacks position information required by the current filter function. This is a limitation we are actively working to address.
> > > > > >
> > > > > > Although the original MaveDB data was not mapped to the reference genome due to historical reasons, a follow-up publication has provided such mapping [1], along with an [official implementation](https://github.com/ave-dcd/dcd_mapping). This work maps the MaveDB scoreset data to GA4GH Variation Representation Specification (VRS) objects. We are planning to replace the current implementation (which directly uses the MaveDB scoreset data) with this mapped version, which will better align MaveDB with the genetic variants (GVs) from other databases.
> > > > > >
> > > > > > Once MaveDB is updated to the mapped version, the filter function will be directly applicable. However, this change has not yet been integrated into the codebase. In the meantime, we could implement a simple patch using the labels from MaveDB, but we believe it would be more elegant to unify the GV format across datasets wherever possible.
> > > > > >
> > > > > > [1] Arbesfeld, J.A., Da, E.Y., Stevenson, J.S., Kuzma, K., Paul, A., Farris, T., Capodanno, B.J., Grindstaff, S.B., Riehle, K., Saraiva-Agostinho, N. and Safer, J.F., 2023. Mapping MAVE data for use in human genomics applications. bioRxiv, pp.2023-06.

---

> > > > > > > ### Comment · Reviewer_3LTZ · 2024-08-21
> > > > > > >
> > > > > > > Good to see that we have the same understanding of that. While technically feasible, I'm not perfectly sure about mapping noncoding/regulatory MAVE results to the genome, and using the genomic context in a FM for prediction - it would seem to me that we introduce "fake context", as the enhancer sequence might be the same, but it was assayed in a very different sequence context to obtain the score.
> > > > > > >
> > > > > > > That being said, MaveDB itself provides a coding/noncoding classification - why not use that?

---

> > > > > > > > ### Author Response · Authors · 2024-08-22
> > > > > > > >
> > > > > > > > Thank you for your suggestion. We are happy to add the option to filter coding versus non-coding regions using the MaveDB label. However, this may take some time, as it will require updates to our cached data files.
> > > > > > > >
> > > > > > > > Regarding the concern about fake contexts, we agree this is a valid issue. The mapping in [1] was initially performed using BLAT, which could result in some mapped sequences having different contexts from the original experiments. That said, the mapped data has been integrated into several important analysis tools, including the UCSC Genome Browser (as a track hub) and the Genomics 2 Proteins Portal. For this reason, we believe it is important to include the mapped data, allowing users to decide how to utilize it.
> > > > > > > >
> > > > > > > > We sincerely appreciate your feedback and suggestions.

---

> > > > > > > > > ### Author Response · Authors · 2024-08-28
> > > > > > > > > **Integrating MaveDB Coding/Noncoding Labels**
> > > > > > > > >
> > > > > > > > > We have taken your suggestion and implemented a filter to distinguish between coding and non-coding regions based on MaveDB metadata. This [update](https://github.com/bowang-lab/genomic-FM/pull/53) is now complete in the codebase.  Thank you so much for your insightful feedback on our paper! We hope you found our responses useful. As the discussion period is coming to a close, please feel free to ask any remaining questions you may have. We're happy to provide further clarification!

---

> > > > > > > > ### Author Response · Authors · 2024-08-30
> > > > > > > >
> > > > > > > > Following our discussion on the coding vs. non-coding filter across all sub-datasets in GV-Rep, we have made the necessary updates to the original codebase. Additionally, we will add the extended review of Genomic Assay Prediction Models and the clarification on the source data formats for the Cell Passport and Score Project datasets in the final manuscript. We kindly ask whether you would consider increase the score?

---

> > > > > > > > > ### Comment · Reviewer_3LTZ · 2024-08-31
> > > > > > > > >
> > > > > > > > > Dear authors,
> > > > > > > > > for reasons I don't know I don't have an edit button for my original review. I was hoping this would start working again, but I left a separate comment to the AC that my score should be increased in any case.

---

### Official Review · Reviewer_TKj5 · 2024-07-19
**GV-Rep**

**Rating:** 6
**Confidence:** 3
**Correctness:** Yes
**Clarity:** Yes

**Review:**

Pros:
1. The dataset is unique in its scope and scale, incorporating a broad range of genetic data and annotations.
2. This dataset could significantly impact genetic research by providing a resource that bridges the gap between genomic data availability and actionable genetic insights.
3. The paper is well-organized, with a clear explanation of the dataset's construction, characteristics, and the experimental setup for model testing.

Cons:
1. Table 2 shows the AUROC scores are quite low for the three different tasks on the proposed GV-Rep dataset.
2. Encourage the development of deep learning models or the adaptation of existing ones to better leverage the dataset's complexity.

**Strengths:**

1. The inclusion of diverse genetic variants and experimental contexts makes this dataset an useful resource for researchers across various fields of genetics and personalized medicine.
2. The experimental results provide a critical evaluation of GFMs, which is essential for understanding and improving these models.

**Additional Feedback:**

None

**Documentation:**

Yes

**Limitations:**

Yes

**Opportunities For Improvement:**

1. Future work could focus on developing or adapting GFMs to improve the downstream evaluation performance.

**Relation To Prior Work:**

Yes

**Summary And Contributions:**

The paper introduces a large-scale dataset designed to improve the learning of genetic variant representations in genomic foundation models. This dataset includes over 7 million records, detailed annotations across diverse genetic traits, diseases, and experimental contexts.

---

> ### Author Rebuttal · Authors · 2024-08-15
>
> We appreciate your suggestions regarding the future work. While the focus of this paper is to publish the dataset and encourage future work with more robust evaluations for genetic variant related tasks, we do realize the potential limitation of existing GFMs and the urges to develop new models to address these challenges.

---

### Official Review · Reviewer_gPE3 · 2024-07-22
**GV-Rep: A good work to promote the development of building Large-Scale Dataset for Genetic Variant Representation Learning**

**Rating:** 6
**Confidence:** 3

**Review:**

Pros:

1.The dataset is extensive, well-annotated, and designed to address key challenges in GV interpretation.

2.The analysis of the dataset’s structure and properties is thorough.

3.The experimentation with pre-trained GFMs provides valuable insights into the current capabilities and limitations of these models.

4.The paper is well-organized, with clear sections detailing the dataset construction, analysis, and experimental results.

5.The dataset has the potential to significantly enhance the accuracy of genetic variant interpretation, which is crucial for advancing personalized medicine and improving patient outcomes.

6.In general, this paper is a good work that can promote the development of building Large-Scale Dataset for Genetic Variant Representation Learning.

Cons:

1.The experiments are rich and diverse. But to a certain extent, this paper is lack some significance and robustness analysis and it would be better to include them.

2.The paper could benefit from a more detailed discussion on the potential negative societal impacts of the work.

3.The key motivation and novelty of this paper should be discussed more.

**Strengths:**

1.The dataset is extensive, well-annotated, and designed to address key challenges in GV interpretation.

2.The analysis of the dataset’s structure and properties is thorough.

3.The experimentation with pre-trained GFMs provides valuable insights into the current capabilities and limitations of these models.

4.The paper is well-organized, with clear sections detailing the dataset construction, analysis, and experimental results.

5.The dataset has the potential to significantly enhance the accuracy of genetic variant interpretation, which is crucial for advancing personalized medicine and improving patient outcomes.

6.In general, this paper is a good work that can promote the development of building Large-Scale Dataset for Genetic Variant Representation Learning.

**Additional Feedback:**

1.Future Work: The authors could outline potential future directions, such as improving model performance in complex scenarios and expanding the dataset to include more diverse genetic contexts.

2.User Guide: Providing a comprehensive user guide or tutorial for using the dataset and the associated tools would be highly beneficial for new users.

**Clarity:**

The paper is well-written, with clear explanations of the dataset construction, analysis, and experimental results. The figures and tables are well-designed and effectively support the text.

**Correctness:**

The claims made in the submission are generally correct, and the dataset appears to be constructed in a sound manner. The evaluation methods and experimental design are appropriate and performed correctly.

**Documentation:**

The documentation is sufficient, with detailed explanations of data collection, organization, and intended uses. The authors provide a URL for accessing the dataset, which is crucial for reproducibility.

**Ethics:**

There are no immediate ethical concerns with the paper.

**Limitations:**

In the current version, the authors have clarified the limitations of their work thoroughly. However, the paper should include a more comprehensive discussion on the societal and ethical implications of using such a dataset, particularly in the context of clinical decision-making and potential biases in the data.

**Opportunities For Improvement:**

1.The experiments are rich and diverse. But to a certain extent, this paper is lack some significance and robustness analysis and it would be better to include them.

2.The paper could benefit from a more detailed discussion on the potential negative societal impacts of the work.

3.The key motivation and novelty of this paper should be discussed more.

**Relation To Prior Work:**

The paper provides a clear discussion of how this work differs from and builds upon previous contributions. The comparison with existing datasets and benchmarks is well-articulated.

**Summary And Contributions:**

This paper introduces GV-Rep, a large-scale dataset designed for deep learning models to learn genetic variant (GV) representations. The dataset contains over 7.5 million records, including comprehensive annotations and data from gene knockout tests. The authors aim to address the challenges in GV interpretation by providing a standardized and richly annotated dataset that can enhance the performance of genomic foundation models (GFMs).

---

> ### Author Rebuttal · Authors · 2024-08-15
>
> We appreciate your feedback on the paper. Below, we provide detailed responses to each of your questions.
>
> ### Discussion on the negative social impact and solutions
>
> While **GV-Rep** is unlikely to have a negative impact on the machine learning community, caution must be exercised whenever this dataset is used to inform clinical decision-making. GV-Rep is derived from a variety of diverse data sources, and there is a potential for bias in these data. When evaluating variant indexing and scaling laws, we performed experiments on cancer-related tasks, however, results can vary widely across different diseases. As discussed in line 284 of the original paper, incorporating sensitive attributes such as ethnicity and sex may help mitigate these biases. Moreover, any applications developed using this dataset should be implemented with care to uphold the principle of human-in-the-loop for real-world applications. It is crucial that all genetic variants predicted by machine learning models are thoroughly reviewed and validated by clinicians to ensure accurate and responsible use in informing patient care in clinical settings.
>
> Additionally, the use of genetic data presents risks to the privacy of an individual's genetic information which can be used for genetic discrimination. We will be responsible for ongoing monitoring and validation as more data is integrated into GV-Rep, ensuring that the dataset continues to reflect the diversity of populations it aims to serve and minimizing the risk of bias over time. This vigilance will help ensure that GV-Rep remains a valuable tool for advancing personalized medicine while safeguarding against unintended consequences.
>
> ### Motivations of the GV-Rep dataset
>
> As detailed in lines 49 to 53 of the original paper, the motivation for building **GV-Rep** stems from the lack of datasets with sufficient size, diversity, standardized formats and biological or clinical relevance. We further outline the limitations of existing GFM evaluations for variant effect prediction in lines 89-101. However, these models lack standardized and meaningful performance assessments, and tend to simplify this problem into a binary classification: pathogenic variants leading to genetic diseases or benign variants found in healthy populations. Moreover, each study reports a different definition of “healthy population” (i.e. 1000G, gnomAD, use of varying MAF cutoff) and “pathogenic variants” (e.g. classified by Ensembl, ClinVar “P”, ClinVar “LP”). **GV-Rep** is designed to circumvent these issues and foster the next generation of genetic variant analysis tools. A key motivation for this work is our observation that many machine learning researchers are unfamiliar with the essential challenges of genetic variant prediction. Despite the availability of some public databases on genetic variants, these resources are often difficult for machine learning researchers to access or comprehend. This necessitated the formulation of **GV-Rep** as a standardized problem, enabling more researchers to contribute to advancing this important area.
>
>
> ### Significance and robustness
> The four experiments (in section 5) has been conducted for 3 times wtih different random seeds. We believe that the reported results are relatively robust. We aim to cover a wider range of the models in the future work.

---

### Official Review · Reviewer_fiQq · 2024-07-26
**Important contribution to the evaluation of genomic foundation models**

**Rating:** 9
**Confidence:** 4
**Correctness:** yes
**Clarity:** yes

**Review:**

This paper presents an important benchmark and dataset that will push the field of genomic foundation models further. The work is clear and original.
- the dataset is constructed to cover the entire genome, have relevance to disease ideniftication, and hav various tasks associated with the data
- They evaluate the benchmarks on many foundation models and demonstrate that they are far from saturating their performance
- provide easy way to access to the benchmark

**Strengths:**

The work is very relevant to the community as it enables the evaluation of different foundation models and test how useful can they be in real life problems.

**Additional Feedback:**

none

**Documentation:**

yes

**Ethics:**

no ethical concerns

**Limitations:**

yes

**Opportunities For Improvement:**

Please further proof read the paper.

**Relation To Prior Work:**

yes

**Summary And Contributions:**

The authors present a new large dataset of genetic variants among DNA sequences within a population. These dataset includes various annotations that serve as both course and fine-grain evaluation tasks. Furthermore, the authors evaluate the latest genomic foundation models on these tasks. On many tasks it is seen that the models barely outperform chance and on other they are far from saturating the performance, making them a great candidate for pushing the field forward.

---

> ### Author Rebuttal · Authors · 2024-08-15
>
> We sincerely appreciate your insightful reviews. Following your suggestions, we have further proofread the paper to correct the identified typos. The revised manuscript will be submitted as the final draft.

---

### Author Rebuttal · Authors · 2024-08-15

## Global Response

We appreciate the valuable feedback from all reviewers. Overall, the reviewers agree that our proposed dataset, **GV-Rep**, is well-motivated and acknowledge its comprehensive coverage of the entire genome, supported by detailed documentation and analysis. Notably, most reviewers (fiQq, gPE3, TKj5) highlighted that **GV-Rep** has the potential to advance personalized medicine by bridging the gap between genomic models and actionable genetic insights.

In response to reviewer gPE3's suggestion, we have expanded our discussion on the potential social impact of our dataset, emphasizing the need for caution to avoid inherent biases in clinical decision-making. Additionally, following reviewer 3LTZ's recommendation, we have added a filter function to differentiate between coding and non-coding variants to the sequence extractor, this change has been reflected in our [code repository](https://github.com/bowang-lab/genomic-FM/pull/52). Now the sequence extractor will optionally provide whether a given genetic variant is within protein-encoding region by refering back to the human genome reference GTF file.

---

### Author Response · Authors · 2024-08-28

Dear reviewers,

Thank you for your feedback on our paper! We hope our responses have been clear and helpful. With the discussion period coming to a close, please feel free to ask any questions. We are more than happy to provide further clarification!

---

### Decision · Program_Chairs · 2024-09-26

**Decision:**

Accept (Poster)

**Comment:**

Whereas the average score of this paper in the OpenReview system is 6.5 (Min: 5, Max: 9), the true average score is 7.0 (Min: 6, Max: 9).
This is because Reviewer 3LTZ could not change the score in the rebuttal period for some unknown reason in the OpenReview system.
He/she tried to change the score from 5 to 7 based on the authors' responses in the syste.

All reviewers think that this paper is acceptable.
One reviewer strongly recommends the acceptance.
This reviewer's evaluation is very high: top 15% of accepted papers.
I think that this is an interesting paper including significant contribution.
I recommend the conference organizers to accept this paper.